# Synchronous Presentation of Rare Brain Tumors in Von Hippel–Lindau Syndrome

**DOI:** 10.3390/diagnostics11061005

**Published:** 2021-05-31

**Authors:** Mariachiara Lodi, Antonio Marrazzo, Antonella Cacchione, Marina Macchiaiolo, Antonino Romanzo, Luciano Mastronardi, Francesca Diomedi-Camassei, Alessia Carboni, Andrea Carai, Carlo Gandolfo, Lidia Monti, Angela Mastronuzzi, Giovanna Stefania Colafati

**Affiliations:** 1Department of Paediatric Haematology/Oncology, Cell and Gene Therapy, Bambino Gesù Children’s Hospital, IRCCS, 00165 Rome, Italy; mariachiara.lodi@opbg.net (M.L.); antonella.cacchione@opbg.net (A.C.); angela.mastronuzzi@opbg.net (A.M.); 2Neuroradiology Unit, Department of Imaging, Bambino Gesù Children’s Hospital, IRCCS, 00165 Rome, Italy; antonio.marrazzo@opbg.net (A.M.); alessia.carboni@opbg.net (A.C.); carlo.gandolfo@opbg.net (C.G.); 3Rare Diseases and Medical Genetics Unit, Bambino Gesù Children’s Hospital, IRCCS, 00165 Rome, Italy; marina.macchiaiolo@opbg.net; 4Ophtalmology Unit, Bambino Gesù Children’s Hospital, IRCCS, 00165 Rome, Italy; antonino.romanzo@opbg.net; 5Department of Surgical Specialities, Division of Neurosurgery, San Filippo Neri Hospital/ASL, 1, 00135 Roma, Italy; mastro@tin.it; 6Department of Laboratories, Pathology Unit, Bambino Gesù Children’s Hospital, IRCCS, 00165 Rome, Italy; francesca.diomedi@opbg.net; 7Neurosurgery Unit, Department of Neuroscience and Neurorehabilitation, Bambino Gesù Children’s Hospital, IRCCS, 00165 Rome, Italy; andrea.carai@opbg.net; 8Department of Radiology, Bambino Gesù Children’s Hospital, IRCCS, 00165 Rome, Italy; lidia.monti@opbg.net

**Keywords:** hemangioblastomas, endolymphatic sac tumor, MRI, Von Hippel–Lindau syndrome

## Abstract

Von Hippel–Lindau (VHL) disease is a heritable cancer syndrome in which benign and malignant tumors and/or cysts develop throughout the central nervous system (CNS) and visceral organs. The disease results from mutations in the VHL tumor suppressor gene located on chromosome 3 (3p25-26). A majority of individuals (60–80%) with VHL disease will develop CNS hemangioblastomas (HMG). Endolymphatic sac tumor (ELST) is an uncommon, locally aggressive tumor located in the medial and posterior petrosal bone region. Its diagnosis is based on clinical, radiological, and pathological correlation, and it can occur in the setting of VHL in up to 10–15% of individuals. We describe a 17-year-old male who presented with a chief complaint of hearing loss. Brain and spine Magnetic Resonance Imaging documented the presence of an expansive lesion in the left cerebellar hemisphere, compatible with HMG in association with a second cerebellopontine lesion compatible with ELST. The peculiarity of the reported case is due to the simultaneous presence of two typical characteristics of VHL, which led to performing comprehensive genetic testing, thus allowing for the diagnosis of VHL. Furthermore, ELST is rare before the fourth decade of life. Early detection of these tumors plays a key role in the optimal management of this condition.

## 1. Introduction

Von Hippel–Lindau syndrome (VHLs) is a rare, inherited disease with autosomal dominant inheritance, caused by the mutation of the VHL gene, and in about 20% of cases the mutation occurs de novo. The incidence of VHLs is about 1/36,000 in the general population and typically manifests in the second decade of life. In the CNS, characteristic HMGs are seen in the retina, brain, and spine with additional ELST. Outside of the CNS, VHL disease presents with tumors of the pacreas, kidneys, adrenals, and reproductive organs. Therefore, it is essential that radiologists be familiar with their imaging appearance [1,2,3].

## 2. Case Report

A 17-year-old male was referred to our hospital with history of left hearing loss associated with visual loss and worsening headache for about a year. Clinical evaluation showed severe left hearing loss and diplopia occurring with the gaze toward the lower quadrants. No additional neurological deficits were detected. No significant familial pathological history was reported. Audiometry confirmed a severe left sensorineural hypacusia.

Brain and spine Magnetic Resonance Imaging (MRI) documented the presence of a left, solid cystic cerebellar mass with inhomogeneous signal intensity and vivid contrast enhancement of the solid component (Figure 1a–d).

Moreover, an additional left cerebellopontine angle (CPA) tumor was detected, extending into the internal auditory canal and incorporating the ipsilateral facial acoustic nerves (Figure 1a–d). A brain CT scan was also performed to allow for a more in-depth pre-operative study (Figure 1e,f).

Ophthalmologic evaluation showed the presence of retinal hemangioblastomas (Figure 2d–f) confirmed with TC and MRI studies (Figure 2a–c).

Surgical resection of both lesions was advised and performed through a left retrosigmoid craniotomy. Complete resection of the cerebellar lesion and subtotal resection of the cerebello-pontine angle one were achieved with no post-operative complications (Figure 3a,b).

Histology of the cerebellar lesion revealed a highly vascular tumor consisting of lobules and sheets of epithelioid cells. Many vacuolated clear cells, focal nuclear atypia, and occasional mitoses were present. Immunostains for S-100 and NSE were positive. Proliferation index (anti-Ki67) resulted in about 3%. A diagnosis of cerebellar hemangioblastoma (HMG) was formulated (Figure 4a–d). Regarding the CPA lesion, it consisted of exile papillary fronds lined by cubic epithelium with minimal nuclear atypia. Cytokeratin immunostain was positive. These findings were consistent with endolymphatic sac tumor (ELST) (Figure 4e,f).

HMG and ELST, in association with retinal angiomatosis, led to suspect a Von Hippel–Lindau Syndrome (VHL).

In consideration of clinical, neuroradiological, and histological features, blood sample was collected from the patient and genetic tests were carried out in our patient by Multiplex Ligation-dependent probe amplification (MLPA). MLPA analysis to detect large deletions in the VHL gene was carried out using the SALSA P016B VHL probe kit (MRC-Holland, Amsterdam, Netherlands). The kit contains eight probes to the VHL gene (four in exon 1 and two in each of exons 2 and 3). Briefly, 100 ng DNA was denaturated at 98 °C for 5 min, and the MLPA probe cocktail was added to a total volume of 8 μL and allowed to hybridize for 16 h at 60 °C. Following addition of Ligase-65 and ligation at 54 °C for 15 min, the ligase was inactivated at 98 °C for 5 min. PCR primers, dNTP, and polymerase mix were then added and PCR was carried out for 33 cycles of (95 °C for 30 s, 60 °C for 30 s, and 75 °C for 60 s). Products were then analyzed using an ABI PRISM 3130x Genetic Analyzer (Applied Biosystems, acquired from Thermo Fisher Scientific Corporation, Waltham, MA, USA) with ROX-500-labeled internal size standard. Data were generated using GeneMapper (GeneMapper Software Version 4.0, Applied Biosystems, acquired from Thermo Fisher Scientific Corporation, Waltham, MA, USA.

The MLPA test confirmed the constitutional VHL gene mutation c.394delC (p.Gln132Lysfs*27) (NM_000551.3); thus, the finding defines the diagnosis of type I Von Hippel–Lindau Syndrome.

Diagnostic imaging was extended to the whole body. Abdominal ultrasonography showed multiple cysts, as in cystic fibrosis (from 1 to 20 mm) in the head and body of the pancreas.

In the presence of suspicion of VHL disease, the study of the spinal cord should be performed using thin-layer sequences to look for involvement of the spinal cord (Figure 5a,b). Radiological follow-up of these lesions should be frequent in view of their high probability of growth.

One year later, due to evidence of progression of the residual ELST in absence of serviceable hearing on the left side, a complete resection was obtained by a translabyrinthic approach, completely sparing facial nerve function.

## 3. Discussion

Von Hippel–Lindau syndrome (VHLs) is a rare, inherited disease with autosomal dominant inheritance, caused by the mutation of the VHL gene. The mutation causes a loss of function of the VHL tumor suppressor gene, located at the level of the short arm of chromosome 3 (3p25.3), which normally favors the degradation of the hypoxia-inducible factor (HIF) by the proteasome in the presence of normal oxygen levels. The lack of oxygen-dependent degradation of HIF leads to the translocation of this factor at the nuclear level and the subsequent transcription of angiogenic, erythropoietic, and growth factors (VEGF, Glut1, PDGF, and TGFα), whose activation determines tumor genesis [4]. Subjects affected by VHLs, due to the overproduction of angiogenic and erythropoietic factors, may develop vascular lesions affecting the central nervous system (CNS), including cerebellar, spinal, brainstem, and supratentorial hemangioblastomas. In addition, retinal angiomas and endolymphatic sac tumors are described. At the visceral level, manifestations include liver and renal cysts, clear cell renal cell carcinomas (RCC), pheochromocytomas, pancreatic neuroendocrine tumors (pNET), epididymal, and broad ligament cystadenomas of the uterus [5].

Endolymphatic sac tumors develop in 10–15% of VHL cases and represent a distinctive lesion. The mean age of onset is lower in VHL patients (31.3 years) than in non-VHL ones (52.5 years) [6] and they may be bilateral in 30% of cases [7]. The tumor arises from the duct and the endolymphatic sac, specialized structures of the vestibular aqueduct included in the thickness of the dura mater, and can be locally invasive and erode the adjacent structures such as the semicircular canals and cochlea. Clinical symptoms include hearing loss (84–100%), tinnitus (73–77%), vertigo (62–68%), and facial nerve palsy (8%). Hearing loss may be sudden (secondary to intralabyrinthine hemorrhage) or gradual (usually owing to endolymphatic hydrops) [7]. On CT, endolymphatic sac tumors may show a moth-eaten appearance in the petrous temporal bone related to the “scalloping” of the surrounding bone structures and erosive changes in the vestibular aqueduct, semicircular canals, and cochlea. Central calcific spiculation and posterior rim calcification are commonly seen. Enhanced images show an avidly enhancing tumor [8,9]. At MRI, these tumors may be T1 hyperintense (secondary to hemorrhagic and proteinaceous contents) and T2 hyperintense. Intense enhancement of the solid portions of the tumor is seen after administration of gadolinium. Accurate early diagnosis of endolymphatic sac tumors followed by prompt surgical resection are critical, as they can help prevent or reduce hearing loss and other audio-vestibular symptoms [10]. The mainstay of ELTS management is complete surgical resection, but when it is not feasible, adjuvant radiotherapy or radiosurgery may be indicated [11].

The CNS HMG is the prototype tumor of VHL and may occur in the cerebellum, brainstem, spinal cord, cauda equina, or supratentorial region. In VHL, CNS HMGs most commonly occur in the cerebellum (44–72%), followed by the spinal cord and brainstem [12,13]. HMGs constitute about 1.5% to 2.5% of all neoplasms arising intracranially and 7% to 12% of all posterior fossa neoplasms [14]. Patients with cerebellar HMG may develop gait ataxia, dysmetria, headaches, diplopia, vertigo, and emesis: In many patients, the symptoms may be caused by the cyst or syrinx associated with the tumor rather than the tumor itself. Historically, cerebellar HMGs have been classified into four subtypes: type 1 (5%) is a simple cyst without a macroscopic nodule; type 2 (60%) is a cyst associated with a mural nodule; type 3 (26%) is a solid tumor; and type 4 (9%) is a solid tumor with small internal cysts [7]. CNS HMGs are classified as World Health Organization (WHO) grade 1 tumors. Histologic analysis reveals a vascular capillary network lined with hyperplastic endothelial cells, which are surrounded by pleomorphic vacuolated stromal cells with lipid-rich cytoplasm. Computed tomographic (CT) images show a well-defined homogeneous cyst with an isoattenuating mural nodule on nonenhanced images [15]. Contrast-enhanced CT images show an avidly enhancing mural nodule within the cyst (the so-called cyst with mural nodule appearance). MR imaging features mimic those at CT. The cystic component is T1 hypointense and T2 hyperintense and the mural nodule is T1 hypointense and T2 isointense or hyperintense. After intravenous injection of gadolinium, the nodules show evident enhancement while no enhancement is seen in the cyst wall. The differential diagnosis includes juvenile pilocytic astrocytoma, arachnoid cyst, and metastases. Surgical resection is performed for the symptomatic tumors, which are usually tumors with large associated cysts and/or peritumoral edema.

Detection of retinal HGMs must lead to a complete workup to exclude or confirm the syndrome, especially if they are multiple and bilateral. In more than 80% of cases, retinal HMGs are peripheral and present on ophthalmoscopic examination as highly vascularized solid nodules, possibly associated with surrounding edema. On CT examination they are slightly hyperdense (Figure 2a) compared to the surrounding vitreous and may be calcific [16] on MRI retinal HMGs are hyperintense in T2WI (Figure 2b) and demonstrate a rich enhancement after Gadolinium injection (Figure 2c) [17]. Retinal detachment, cataract, uveitis, macular edema can be can be associated with retinal HGMs due to exudative phenomena [2,16].

Frequently spinal cord lesions coexist with cerebellar ones, which is why it is essential to perform a complete study of the spinal cord as well [12,18]. At the spinal level, HMGs are often located in the cervical portion. Similar to the cerebellar HMGs from radiology and morphology point of view, they can be both intra- and extrame-dullary and tend to show the nodular part facing the posterior pial side [17].

HMGs can be associated with medullary edema or with hydrosyringomyelias due to compressive alteration of the CSF dynamics. The vascular nature, which may also be associated with the presence of intralesional arteriovenous shunts [15,19], can make differential diagnosis difficult, especially with spinal vascular lesions (dural fistulas or arteriovenous malformations).

The clinical diagnosis of VHLs according to Danish criteria contemplate:-family history for VHLs and diagnosis of retinal hemangioblastoma, CNS hemangioblastoma, PPGL, or RCC;-negative family history for VHLs, in the presence of ≥2 CNS or retinal lesions (hemangioblastomas) or ≥2 visceral lesions (RCC, PPGL, multiple pancreatic or renal cysts, pNET, tumor of the endolymphatic sac of the o- cystadenoma of the epididymis, or broad ligament of the uterus) or the concomitant presence of a hemangioblastoma and a visceral lesion [20].

The diagnosis of VHLs may be suspected clinically but confirmed by genetic analysis. The identification of a pathological variant of the VHL gene currently represents the diagnostic gold standard.

Comprehensive genetic testing of the VHL gene is recommended for individuals with the presence of one or more characteristic lesions including hemangioblastomas of the brain, spine, or retina; endolymphatic sac tumors; epididymal cystadenomas; pheochromocytoma; multiple pancreatic cystadenomas; and clear cell RCC diagnosed at age 40 or younger. There is also a large variety of hereditary cancer panels, many of which include the VHL gene. Thus, individuals might be diagnosed with a pathogenic variant in VHL due to other combinations of tumor diagnoses and family history (e.g., those with a diagnosis of clear cell RCC over the age of 40 and a family history of kidney cancer). Early detection of these tumors plays a key role in the optimal management of this condition. Radiologists should be aware of the imaging features of VHLs.

For asymptomatic patients, we suggest observation with serial imaging, unless tumor growth appears or symptoms develop, rather than immediate therapeutic intervention.

This peculiar case had the aim to highlight the importance of having recognized two different rare tumors, allowing us to achieve the genetic diagnosis and to begin an adequate follow-up.

## Figures and Tables

**Figure 1 diagnostics-11-01005-f001:**
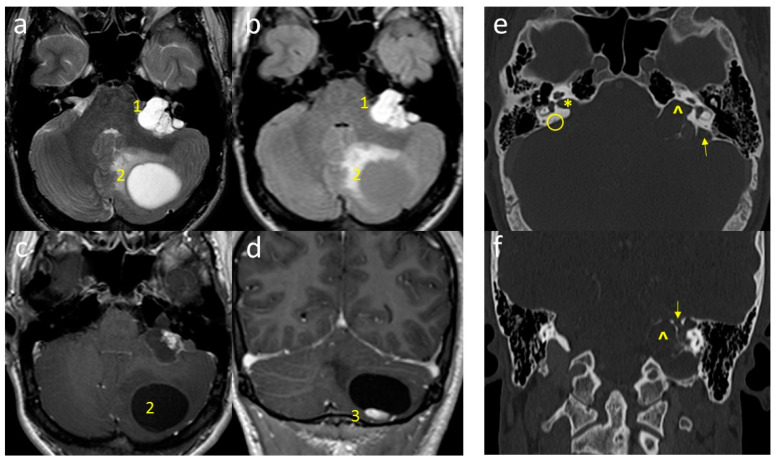
Papillary cystadenoma of the endolymphatic sac and cerebellar hemangioblastoma (HMG) in VHL. (**a**,**b**): Axial T2-WI and FLAIR images; (**c**,**d**): axial and coronal post-contrast T1-WI images. A cystic mass (without signal suppression on FLAIR sequence) grows through the mastoid and temporal bones, disrupting bone structures (1, ELST). After Gadolinium injection, the mass becomes relatively more homogeneous with blunt peripheral rim enhancement. HMG appears as cystic mass (2) in the cerebellar hemisphere, showing isolated and peripheral enhancing mural nodule (3). (**e**,**f**): Axial and coronal computerized tomography (CT) scans show bone destruction of the temporal pyramid, more evident while comparing healthy contralateral. Normal-appearing, right-sided inner ear (star) and endolymphatic duct (circle) are no longer recognizable contralaterally (respectively, arrowhead and thin arrow).

**Figure 2 diagnostics-11-01005-f002:**
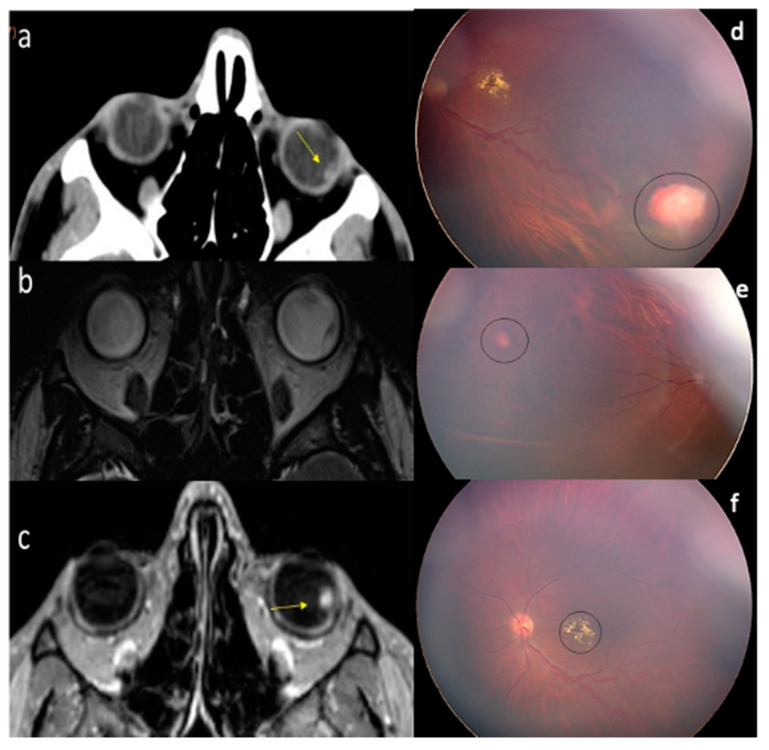
HMG of the retina. (**a**–**c**): Axial CT scan, T2-WI (volumetric sequence) and axial post-contrast T1-WI images. A focal hyperdense and isointense (compared to surrounding vitreous) peripheral nodule is depicted on the lateral portion of the left retina (arrow). A blunt contrast enhancement is typical of this hypervascular tumor. (**d**–**f**): Ophthalmoscopy.

**Figure 3 diagnostics-11-01005-f003:**
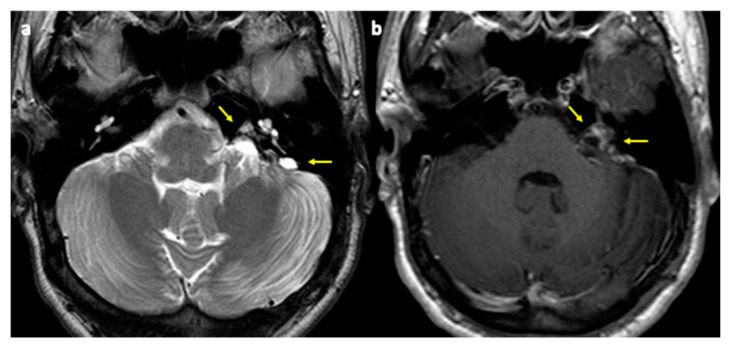
Follow-up MRI. Axial T2w (**a**) and GdT1w (**b**) images show complete surgical resection of the left cerebellar tumor and partial resection of the ipsilateral cerebellum-pontine angle mass with residual cystic appearance (**a**, arrows) and lesional contrast enhancement after gadolinium injection (**b**, arrows).

**Figure 4 diagnostics-11-01005-f004:**
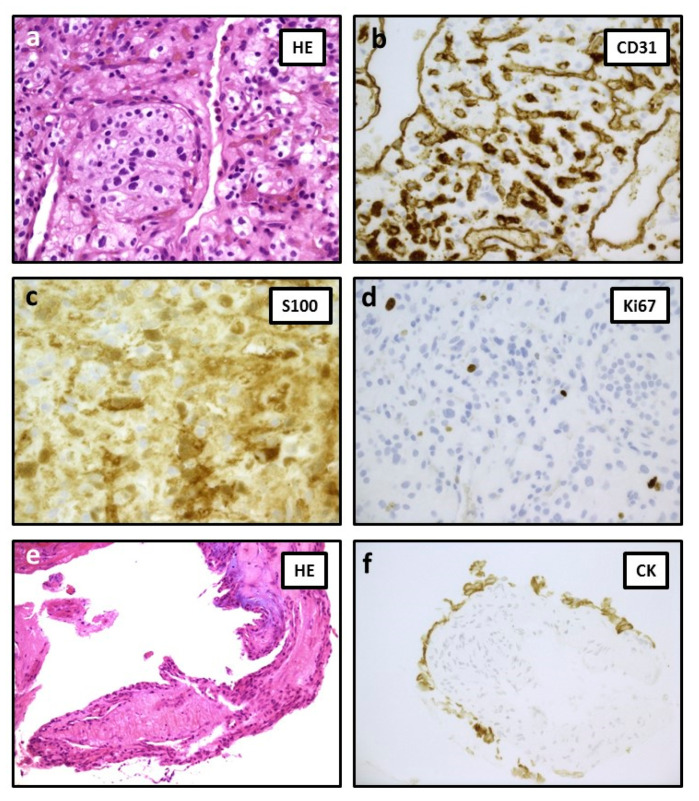
HMG (**a**) highly vascular neoplasia formed by lobules and nets of epithelioid cells, many with clear cytoplasm. Focal nuclear atypia is observed (40×). (**b**) Anti-endothelial marker CD31 revealed a rich intermingled vascularity. Note that neoplastic cells are negative for this vascular marker (40×). (**c**) S100 immunostain resulted as positive in the majority of cells (63×). (**d**) Proliferation rate was about 3% (40×). ELST (**e**) thin papillary fibrovascular core covered by cuboidal epithelium layer (20×). (**f**) Positivity for cytokeratin immunostain (40×).

**Figure 5 diagnostics-11-01005-f005:**
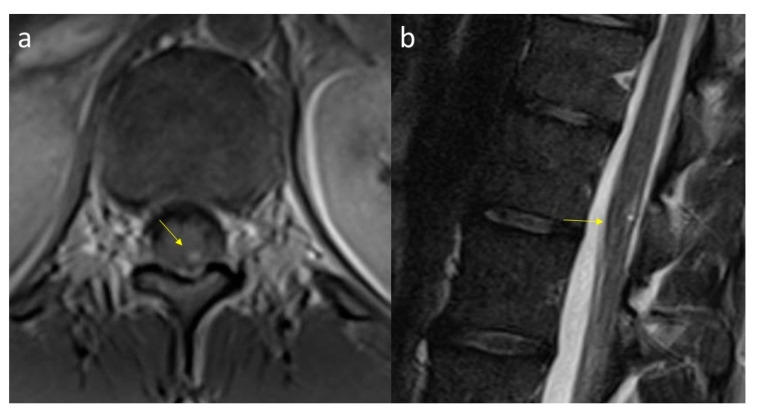
HMG of the spinal cord. Axial post-contrast T1-WI (**a**) and sagittal T2-WI images (**b**). A peripheral, barely visible, hyperintense (**b**), and enhancing nodule (**a**) is depicted on the posterior edge of the conus medullaris (arrows).

## Data Availability

The data presented in this study are available on request from the corresponding author.

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
