# Peer review of "Synchronous Presentation of Rare Brain Tumors in Von Hippel–Lindau Syndrome"

_diagnostics, 2021, doi:10.3390/diagnostics11061005_

Round 1
Reviewer 1 Report
Nicely presented report!
Lines 169 - 184 are superfluous and may be removed.
Please add post-op images following cranial surgeries.
Please add histopathology images from hemangioblastoma and ELST.
Was an ophthalmology exam performed? Fundcoscopy results could be reported in text and as a panel in Figure 2.
Figures 3 and 4 do not add much to the report.
Author Response
.
Reviewer 2 Report
The authors present the investigation of a patient who initially presents with an endolymphatic sac tumour and is eventually diagnosed with Von Hippel Lindau disease. ELST are a more uncommon manifestation of VHL disease, and this article brings attention this as a portal of VHL diagnosis.
The article is well written from the radiologic and clinical perspective. Some improvements include:
- The authors suggest a thorough workup of a hemangioblastoma is merited (line 131). This is true. The converse for ELST should also be mentioned. Should all ELST have a VHL workup?
- The genetic section (line 185) seems separate, and could be split to bring up some of the background of the genetics of VHL in the introduction.
- The VHL variant in c. and p. along with NM number and the interpretation should be added. The subtype of VHL should be also added (VHL type 1, 2a, 2b, and 2c).
- The diagnostic criteria for VHL disease varies (Danish criteria, VHL Alliance etc). This should be mentioned.
- Minor typos (ELST, pathological variant)
Author Response
Comments to Author
Review -1185267
1. The authors suggest a thorough workup of a hemangioblastoma is merited (line 131). This is true. The converse for ELST should also be mentioned. Should all ELST have a VHL workup?
Although these tumors also occur sporadically, ELSTs are common in patients with VHL disease, within an incidence of approximately 15 percent on detailed evaluation. In one series, for example, bilateral tumors were present in 28 percent of the patients with VHL versus 1 percent in the patients without VHL disease.
